# *Aspergillus* in Critically Ill COVID-19 Patients: A Scoping Review

**DOI:** 10.3390/jcm10112469

**Published:** 2021-06-02

**Authors:** Erlangga Yusuf, Leonard Seghers, Rogier A. S. Hoek, Johannes P. C. van den Akker, Lonneke G. M. Bode, Bart J. A. Rijnders

**Affiliations:** 1Department of Medical Microbiology and Infectious Disease, Erasmus University Medical Center, 3015 GD Rotterdam, The Netherlands; l.bode@erasmusmc.nl (L.G.M.B.); b.rijnders@erasmusmc.nl (B.J.A.R.); 2Department of Pulmonary Medicine, Erasmus University Medical Center, 3015 GD Rotterdam, The Netherlands; l.seghers@erasmusmc.nl (L.S.); r.hoek@erasmusmc.nl (R.A.S.H.); 3Department of Intensive Care Medicine, Erasmus University Medical Center, 3015 GD Rotterdam, The Netherlands; j.vandenakker@erasmusmc.nl; 4Department of Internal Medicine, Erasmus University Medical Center, 3015 GD Rotterdam, The Netherlands

**Keywords:** COVID-19, *Aspergillus*, scoping review, mortality

## Abstract

Several reports have been published on *Aspergillus* findings in COVID-19 patients leading to a proposition of new disease entity COVID-19-associated pulmonary aspergillosis. This scoping review is designed at clarifying the concepts on how the findings of *Aspergillus* spp. in COVID-19 patients were interpreted. We searched Medline to identify the studies on *Aspergillus* spp. findings in COVID-19 patients. Included were observational studies containing the following information: explicit mention of the total number of the study population, study period, reason for obtaining respiratory samples, case definition, and clinical outcomes. Excluded were case series, case reports and reviews. Identified were 123 publications, and 8 observational studies were included. From the included studies the following issues were identified. The proportion of immunocompromised patients considered as host factors varied from 0 to 17%. Most of the studies did not mention radiographic findings explicitly. Respiratory samples were mostly obtained to investigate clinical deterioration. *Aspergillus* culture, antigen or PCR testing on bronchoalveolar lavage (BAL) fluid were performed in between 23.3% and 66.3% of the study population. Two studies performed periodic samples of BAL. Galactomannan index (GI) positivity in BAL was between 10% and 28%. GI in blood was found in 0.9% to 6.7% of the available samples. The prevalence of COVID-19-associated pulmonary aspergillosis ranged from 2.7% to 27.7%. Studies compared the mortality between defined cases and non-cases, and all showed increased mortality in cases. No studies showed that antifungal treatment reduced mortality. Concluding, this review showed how studies defined the clinical entity COVID-19-associated pulmonary aspergillosis where positive *Aspergillus* test in the respiratory sample was the main driver for the diagnosis. There were many differences between studies in terms of test algorithm and *Aspergillus* test used that largely determined the prevalence. Whether antifungal therapy, either as prophylaxis, pre-emptive or targeted therapy will lead to better outcomes of COVID-19-associated pulmonary aspergillosis patients is still need to be answered.

## 1. Introduction

*Aspergillus* spp. are ubiquitous in the environment and can colonize the human respiratory tract. In immunocompromised patients such as those with prolonged neutropenia, the use of immunosuppressive agents, or those admitted to the intensive care unit (ICU) with severe influenza, colonization can lead to invasive pulmonary aspergillosis (IPA) and tracheobronchitis that is associated with high mortality [1,2]. Soon after the start of the COVID-19 pandemic, clinicians reported a high incidence of *Aspergillus* findings, i.e., positive *Aspergillus* culture or positive galactomannan index from any type of respiratory sample in patients admitted to the ICU with COVID-19. It started with case reports or small case series, that subsequently were summarized in reviews [3,4].

The case reports describe high mortality but are prone to publication bias, and it remains to be shown convincingly that *Aspergillus*-associated mortality in COVID-19 patients is higher compared to well-matched COVID-19 patients that tested negative for *Aspergillus*. Such information might be available from observational studies. Moreover, from large observational studies, more standardized and structured information could be obtained to estimate the importance of positive mycological findings in patients with COVID-19 admitted to the ICU.

This study is a scoping review that by its nature is designed at identifying knowledge gaps, scope the body of literature, and clarify concepts [5]. The aim of this study was to review how the findings of *Aspergillus* spp. in COVID-19 patients were interpreted, and whether it was associated with increased mortality. This scoping review differs from the published systematic reviews [6,7] that used an entity COVID-19-associated pulmonary aspergillosis as starting point.

## 2. Material and Methods

### 2.1. Search Strategy

We searched Medline up to 21 March 2021 for studies on *Aspergillus* and COVID-19 using search terms described in Appendix A No language restriction was applied. No additional databases were searched because this review was not aimed to pool the entire and rapidly growing published literature (due to possible publication bias), but to find an explanation of the high incidence of positive *Aspergillus* tests (culture, galactomannan, PCR, other) in COVID-19 patients in the ICU. Therefore, studies only available in pre-print or were not peer-reviewed were also not included.

### 2.2. Inclusion and Exclusion Criteria

The titles and the abstracts of the included studies were screened by three medical students (SK, SY, and CD), and one of the authors (EY, a clinical microbiologist and epidemiologist) for obvious exclusions, and full texts of the remaining studies were read to find appropriate studies. Included were observational studies with a defined study population (arbitrarily set at >35 patients at the initiation of this review), i.e., cohort studies or cross-sectional studies that explicitly mention the total number of the study population in a certain setting, the study period, the information on why and how respiratory samples were obtained, case definition, and clinical outcomes. Excluded were case series, case reports and reviews.

### 2.3. Data Extraction

We extracted the following information: study setting (geographical location, tertiary/academic or non-academic hospitals, number of patients and number of those with mechanical ventilation), demographic characteristics of the study population, immunocompromised conditions, reason to obtain respiratory samples, use of corticosteroids and antibiotics, radiological findings, type of respiratory samples, microbiological tests (culture, galactomannan or other serological tests, or PCR), case definition, and effect size and treatment.

### 2.4. Assessment of Study Quality

Study quality of included observational studies was assessed using an adapted Newcastle–Ottawa score [8] by SK, SY, and CD in consensus, and reviewed by EY who solved eventual discrepancies. The Newcastle–Ottawa score assessed selection, comparability and outcomes domains, and studies could be categorized as good, fair, or poor. Adaptation was needed so this score could be used specifically for the present study (Appendix A).

## 3. Results

### 3.1. Literature Flow and Study Quality Assessment

The literature flow of observational studies is shown in Figure 1. From 123 publications identified, eight observational studies were included after the selection process. Using Newcastle–Ottawa score, three studies were classified as good [9,10,11] (Appendix A Appendix A). Two studies were deemed as fair [12,13], since they did not assess the outcomes (mortality) in all included patients. One study assessed mortality only in patients of whom respiratory samples were available [12] and another mentioned exclusion of patients without a microbiological diagnosis of fungal infection [13]. Three studies were deemed as poor quality in term of methodology, since the study cohort was not clearly described [14] or not representative, i.e., analysis was performed only in patients with respiratory samples available [15,16].

### 3.2. Characteristics of the Studies and Included Patients

The studies originated from France [9,10,11,15,16], Spain [13], Wales [14], and Italy [12]. The number of included ICU patients ranged from 45 [9], performed in a single center in France, to 260 [11]. All studies were performed in university or tertiary hospitals, except one that was performed in a general hospital [16] and one countrywide study based on samples sent to a national public health institute in Wales [14] (Table 1). The median age of the included patients varied from 57 [14] to 65-years-old [16], and they were mostly male (71.1% [9] to 88% [15]).

### 3.3. Host Factors and Radiological Criterion

Various case definition criteria exist and have been used in interpreting the significance of *Aspergillus* findings in COVID-19 patients [17,18,19,20]. Some of them were inspired by *Aspergillus* findings in influenza patients. These criteria are reviewed in [21]. In general, the criteria need a host factor, compatible clinical or radiological criteria, and mycological criteria. The extracted information regarding the immunocompromised status is presented in Table 1. Other extracted data are presented in Table 2.

The proportion of immunocompromised patients considered as host factors in some case definitions varied from 0 [13] to 17.6% [12]. It should be noted, however, that the denominators calculated by the authors of these publications were not representing the entire COVID-19 ICU patient population. Calculations were either based on patients with available respiratory samples only [12,15], or the data on the immunocompromised status of the patient were exclusively available from patients with *Aspergillus* findings [13]. Data on corticosteroid use were available from all studies except [9] and [16], and typically the studies mentioned the proportion of patients with positive *Aspergillus* tests who received corticosteroids (ranged from 28.6% [15] to 60% [12]).

To summarize data regarding host factors, there was a wide variation in the number of immunocompromised patients in the included studies, with the highest reported percentage of 17.6%. The interpretation is further complicated due to variation in the calculation.

Most of the studies [9,11,12,15] did not mention radiographic findings explicitly, perhaps because a CT at the time *Aspergillus* was detected was not available, while a bedside chest X-ray is not very useful to detect possibly more specific radiological findings. Radiographic findings that have been considered specific in immunocompromised patients with pulmonary aspergillosis like cavitary lesions were mentioned in 1.1% [10] to 1.9% of the patients [14].

To summarize radiological data, the number of specific radiological findings in COVID-19 patients in whom an *Aspergillus* test was positive, was very low (<2%).

### 3.4. Mycological Criterion

Respiratory samples were obtained as routine periodical sampling [9,12,13], to investigate clinical deterioration [10,11,14,15], or when weekly tests of serum galactomannan or (1-3)-β-d-glucanwere positive [16]. *Aspergillus* culture, antigen or PCR testing on BAL fluid were important criteria in all published case definitions [17,18,19,20] and were performed with a variable frequency of 23.3% [14] and 66.3% [12] of the ICU patients (Table 3). No information was available on the number of BAL in two studies [9,13]. Two studies performed periodic samples of BAL, one with a median number of 2 samples per patient (interquartile range 2 to 5) [12] and the other with a mean number of 3 samples per patient [11]. Galactomannan index positivity in BAL fluid varied substantially. It was 10% in one study [15] and around 28% in two studies, one with periodic BAL and the other in the study with “non-directed” BAL [12,14]. Interestingly the proportion of galactomannan index positive BAL decreased from 30% to 8% and 5% during follow-up in the study with periodic BAL sampling [12] but no explanation was given. One study mentioned the proportion of positive galactomannan index among all BAL samples instead of the proportion among patients, where 11 out of 333 (3.3%) BAL samples from 145 patients were positive [11]. Galactomannan index positivity in BAL [12] and other mycological criteria, such as positive culture [10,15], or positive *Aspergillus* PCR [9,14] in BAL fluid were the main driver of case definition as can be seen in the number of positivity of mycological criteria that was almost similar with the number of cases (Table 3).

Galactomannan index in blood, which can be used as a marker for invasive diseases, was found in 0.9% [12] to 6.7% of the available samples [9]. In one study [14], *Aspergillus* unspecific serum biomarkers (1-3)-β-d-glucan was used and the positive rate was as high as 15.6%.

To summarize mycological data, the included studies frequently reported the use of galactomannan index and culture in BAL samples of COVID-19 patients. Often, a high proportion of patients with positive galactomannan and PCR in BAL was reported (up to 28% and 33%, respectively). The proportion of positive *Aspergillus* spp. found in culture was lower than detected using galactomannan. Sign of invasiveness as measured by galactomannan index in serum was rarely found. In the end, the mycological criteria determined the clinical entity.

### 3.5. Ascertaining Clinical Significance to Positive Aspergillus Findings

The prevalence of positive *Aspergillus* tests that were considered to be a case of IPA according to the criteria used by each of the individual studies ranged from 2.7% [11] to 33.3% [9]. In the latter study, mycological criteria included *Aspergillus* PCR positive from any respiratory sample. When a stricter set of criteria was used (i.e., modified AspICU [21] to which a positive PCR was added [9,20]) the prevalence was 15.6%. Several studies compared the mortality between IPA-defined cases and non-cases, and all showed an increased mortality in cases, i.e., 44% vs. 19% [12], 71.4% vs. 36.8% [15], 28.6% vs. 13.3% [9], 86% vs. 37% [13], and 47.1% vs. 31.3% [14]. The comparison was performed against patients with respiratory samples available and not against other COVID-19 patients admitted to the ICU. All studies except [10,16] tried to evaluate whether being a case was independently associated with mortality. Some but not all studies describe the antifungal treatment of the cases. Overall mortality was high despite antifungal therapy. In a study, 43% of the secondary *Aspergillus* cases were treated, and 46% of these treated patients died [12]. In two other studies, 100% mortality was shown among 32% [15] and among 13% [9] of patients with positive *Aspergillus* tests who were treated with antifungal therapy. The reasons why some were treated, while others were not reported.

To summarize mortality and treatment data, it seems that COVID-19 patients in whom *Aspergillus* spp. was found in the respiratory samples had higher mortality risk than in those without *Aspergillus* spp. in their samples. Yet, antifungal therapy did not change this outcome.

## 4. Discussion

This scoping review helps to understand how positive diagnostic tests for *Aspergillus,* whether a positive culture, antigen test or PCR, in critically ill COVID-19 patients are interpreted. In general, the COVID-19 patients were ascertained to a clinical entity IPA merely based on positive *Aspergillus* spp. in lower respiratory samples (the link was almost one-to-one, i.e., every patient who fulfilled a mycological criterion was deemed as a case) and no sign of invasiveness in blood was found in almost all of these patients. Radiology could not differentiate the IPA versus non IPA in COVID-19 patients. Regarding a concurrent host factor, only a small part of patients categorized as having IPA were also immunocompromised.

It could be expected that the microbiological criterion is paramount because no additional host factors are included in any of the diagnostic criteria that are in use, and most critically ill patients fulfill the radiological criterion by having COVID-19 related opacities. These opacities would interfere with the radiologic hallmarks of invasive fungal infections. A critical look at how the patient samples were collected on which these tests were performed, and whether any positive test results were correlated with proven IPA is therefore essential. Lower respiratory tract samples are the best specimens for fungal diagnostics, but their use is limited due to the risk of COVID-19 transmission to healthcare workers. The studies reporting the highest prevalence of the proposed clinical entity COVID-19-associated pulmonary aspergillosis (CAPA) were typically studies reporting positive *Aspergillus* test (culture or PCR) in any type of respiratory samples, including suboptimal lower respiratory tract sampling. A positive *Aspergillus* test in an upper respiratory tract sample is no proof of infection but could be a trigger to look for evidence for lower respiratory tract infection. A study that reported the high prevalence of *Aspergillus* findings was based on the high prevalence of positive galactomannan index in BAL [12]. Interestingly, in the same study, the prevalence of positive galactomannan index decreased during follow-up without explanation given. It is possible that the decrease is due to the treatment effect, but it is also possible that the galactomannan index was not stable. The galactomannan index in a BAL can be expected to depend on several factors, such as the volume used to perform the procedure, and the method of sampling (e.g., CT-guided directed versus bronchoscopic non-directed BAL versus a non-bronchoscopic blind lavage). Given the severity of the lung disease of COVID-19 patients and small respiratory margins of patients on mechanical ventilation, small BAL volumes are typically used (e.g., 20 mL) rather than the 50 to 100 mL more typically done in haematology patients with a suspected IPA. Interestingly, in ventilator-associated pneumonia, smaller BAL volume was not shown to be associated with mortality [22], but a smaller volume results in less dilution of the antigen that is tested a higher cut-off may need to be used. The authors of a proposed working definition of CAPA recognize this problem [21]. Indeed, a very high galactomannan cut-off was suggested for non-bronchoscopic lavage but unfortunately without evidence in support of this high cut-off except for the fact that higher cut-offs typically lead to a more specific test result and a higher positive predictive value. Finally, even what is called a bronchoscopic lavage in this guideline (as opposed to a non-bronchoscopic lavage) cannot automatically be considered one and the same procedure. One difference is whether or not the scope is wedged or not before the lavage is done. Aseptic technique, bronchoscope contamination, use of antibiotics, and preparation of BAL samples at laboratory level during pandemics are may also cause positive galactomannan index without associated COVID-19 pathophysiology. None of the included studies discussed these technical aspects.

Since the prevalence of positive *Aspergillus* tests is mainly determined by mycological test, it is important to look in-depth at the test strategy in the studies. For epidemiological purposes, data originating from regular sampling of respiratory samples of all ICU patients can be used where a variation between 3.3% [13] and 27% [12] was shown. The respiratory samples were also performed in patients who were deteriorated. The most common reason to take respiratory samples was indeed to search for secondary infection in patients who were clinically deteriorated, and these studies showed prevalence between 2.7% [11] and 8% [10,15] among all ICU patients. One study showed a different approach where respiratory samples were taken only in serologic tests for fungi where a prevalence of 3.7% was shown among all ICU patients [16]. From these observations, we can conclude that the variation in the prevalence of secondary invasive Aspergillosis in COVID-19 patients was partly determined by the testing strategy. The prevalence of positive *Aspergillus* spp. in respiratory samples of COVID-19 patients was high when there was a screening for the presence of *Aspergillus* spp. There is another hypothetical explanation why the prevalence of invasive Aspergillosis in COVID-19 patients was high, i.e., the use of immunomodulatory therapies such as corticosteroids. Not all included studies reported the use of corticosteroids, and there is a large variation in the proportion of patients receiving corticosteroids. No study has investigated the association between the use of corticosteroids and other immunomodulatory therapies such as anti-IL-6 Tocilizumab and secondary invasive *Aspergillosis* in COVID-19 patients. A study investigated the association between the corticosteroid and (1-3)-β-d-glucan. It showed that the use of corticosteroid was associated with a higher risk of having multiple positive (1-3)-β-d-glucan test (odds ratio of 7.9 (95% CI: 1.6–39.3) in a small proportion of the study population with corticosteroids data available [14]. 

In some settings, the number of positive *Aspergillus* tests that were considered to be a case of CAPA appears to be low. Next to the abovementioned possible reasons, other factors that might play role in the variation in prevalence are difficulties in making the diagnosis when radiographic findings are not specific, and serum galactomannan is not sensitive.

In the included studies, documented tissue invasion of *Aspergillus* was very rarely documented, as illustrated by the negative serum galactomannan test results in almost all patients, including in those with proven CAPA cases. A study that used another serology assay, i.e., (1-3)-β-d-glucan showed a relatively high proportion of this marker in the blood of COVID-19 patients [14]. The increased number of patients with the positive test, does not imply the increased sensitivity of this marker, it is rather that this assay is a nonspecific marker for *Aspergillus.* In an autopsy study, none of the lung biopsies showed any presence of invasive aspergillosis evidence despite in this study high galactomannan index (>1) in BAL fluid in CAPA patients were found [23].

We further noted that positive *Aspergillus* test results in COVID-19 patients were associated with a significantly higher overall mortality. Only in one study with low-quality assessment using Newcastle–Ottawa score [9] mortality of <30% was shown in COVID-19 patients with secondary invasive aspergillosis. Yet, despite antifungal therapy, the mortality remained high. Until better studies correct for differences in disease severity in those with and without *Aspergillus*, the only conclusion that we tend to draw regarding this high mortality is that *Aspergillus* is a marker of disease severity. Several studies describe high mortality despite antifungal therapy. There are several possible reasons why the higher mortality may be severely confounded. First, very few studies tried to distillate the attributable mortality of CAPA. The arguably most convincing way to estimate the attributable mortality of *Aspergillus*, if any, would be to randomize thousands of ICU patients with COVID-19 to a trial in which antifungal prophylaxis is evaluated. If the incidence of *Aspergillus* drops substantially but no decrease in mortality is observed, the attributable mortality will be minimal. In the study by Bartoletti, et al. [12] CAPA was a risk factor for mortality in a logistic regression model that corrected for SOFA score, age, sex and renal replacement therapy at ICU admission. However, in contrast to IPA in patients with influenza, CAPA is typically diagnosed in the second week after ICU admission and in a significant proportion even later. This means that analysis with the goal to distinguish the attributable from the overall mortality of patients with CAPA, risk factors for mortality to be included in the statistical modal should be those present at the time of the BAL rather than at ICU admission. This could be done by matching CAPA cases in which a positive BAL was the mycological marker, with control patients with negative mycological markers on BAL. They should be matched for SOFA and duration of ICU stay and perhaps also severity of acute respiratory distress syndrome (ARDS) at the time of BAL sampling. The mortality difference that persists between cases and controls in this analysis can inform us about the possible effects of antifungal therapy for CAPA. Second, publication bias should also be considered. Indeed, it is unlikely that, during the hectic of the first months of a pandemic, clinicians will put great effort in publishing that they found a low incidence of *Aspergillus* in COVID-19. More recently, these studies are becoming available as well [11,24,25]. We found the prevalence of positive *Aspergillus* test in COVID-19 patients to be comparable to patients with pneumococcal pneumonia (around 5%) but substantially lower than in patients with influenza (around 19%) among ICU patients (25). This finding supports the findings from Razazi and co-workers who showed that the number of probable invasive *Aspergillosis* in COVID-19 patients with ARDS (C-ARDS) was lower than in the group of ARDS patients due to other viruses (NC-ARDS) (10). In their population, C-ARDS patients more often required neuromuscular blockade, prone positioning, nitric oxide inhalation, extra-corporeal membrane oxygenation support, and longer duration of mechanical ventilation than NC-ARDS patients. The in-ICU and day 28 mortalities were similar in both groups. To date, none of the published studies originated from the second COVID-19 wave. There is a possibility that it is more difficult to publish findings that are not more novel (publication bias). We can also hypothesize that the improved care of COVID-19 patients may contribute to a lower risk of encountering *Aspergillus*. During the second wave, hospitals are more prepared unlike the first wave where temporary facilities in hospitals with a lack of rigorous ventilation requirements were used, which may increase fungal exposure.

Interestingly, while case reports on positive *Aspergillus* tests in COVID-19 patients originate from 17 different countries [26], observational studies are mostly from France. There is a possibility that *Aspergillus* testing, especially the use of galactomannan, is more widespread in French ICUs than in other countries.

It is very much a question of whether antifungal therapy will lead to better outcomes of COVID-19 patients with positive *Aspergillus* tests in the respiratory samples. Options for antifungal treatment in patients with proven invasive Aspergillosis are voriconazole or isavuconazole [21]. In some settings where voriconazole resistance is present, a double therapy, for example, by adding echinocandin, may be considered. Due to the possible high prevalence of *Aspergillus* spp. another question that might be posed is regarding the use of antifungal agents as prophylactic or pre-emptive therapy. There is, for example, an ongoing trial on isavuconazole for the prevention of CAPA (ClinicalTrials.gov Identifier: NCT04707703). In our center, amphotericin B inhalation is used when a positive galactomannan index is found in the deep respiratory sample without positive PCR or culture.

The incidence of *Aspergillus* finding in COVID-19 patients according to observational studies is strikingly higher compared with published autopsy series. A recent systematic review of autopsy studies in which 702 COVID-19 patients were included (https://www.medrxiv.org/content/10.1101/2021.01.13.21249761v1, accessed on 22 March 2021) showed evidence of invasive mould infection in 1.6%. We agree that a very prolonged antifungal therapy preceding the death of a patient with CAPA could lead to a false negative autopsy but it will not explain the huge gap between the incidence of CAPA at autopsy and during ICU stay.

In conclusion, in this scoping review, we identified several factors that should be considered in determining the significance of *Aspergillus* spp. findings in respiratory materials of COVID-19 patients, for clinical and research purposes.

## Figures and Tables

**Figure 1 jcm-10-02469-f001:**
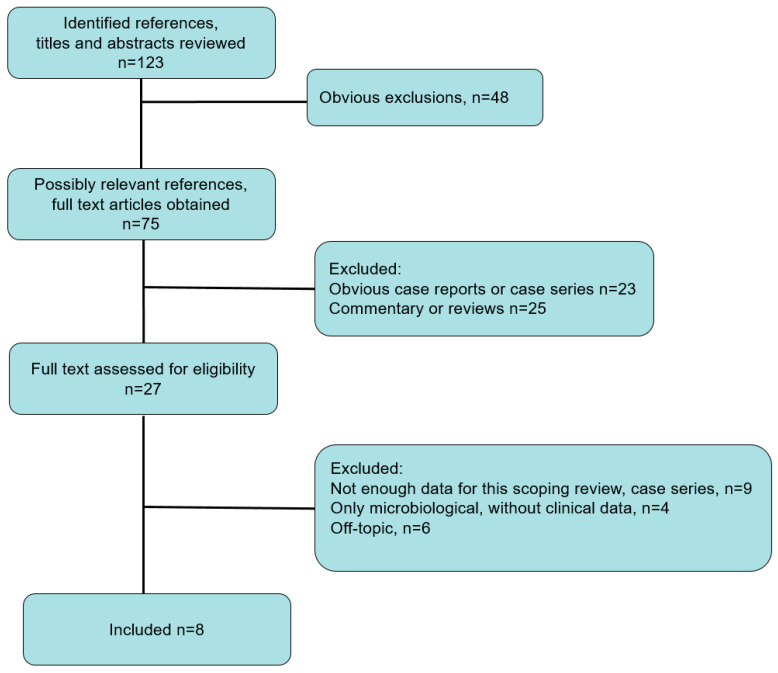
Literature flow.

**Table 1 jcm-10-02469-t001:** Demographic and clinical characteristics of the included studies (cases: what later deemed to fulfil the criteria for the clinical entity, non-cases: not fulfilled clinical entity criteria), ordered by alphabetical order of first author’s names.

First Author (ref)	Setting	Demographic Characteristics	ImmunocomproMised Conditions, n/N (%)	Reason to Obtain Respiratory Samples	Immunomodulatory Drugs Use, n/N (%)	Antibiotics, n/N (%)
Bartoletti [12]	Bologna, Italy 22 February to 20 April 2020 4 ICUs of 3 university hospitals N = 163 mechanically ventilated patients (108 were screened for *Aspergillus*)	Median age (IQR): 64 (57 to 70) years, 78% male Baseline SOFA score, median (IQR): 4 (3 to 5) Prone position, *n* = 22 (76%) in cases, vs. 56 (83%) in non-cases	19/108 (17.6) Malignancies: *n* = 7, Solid organ transplant: *n* = 5 Chronic steroid: *n* = 7	Standard BAL every on day 0–2, 7–9 or clinical worsening	Corticosteroids: In cases: 18/30 (60) vs. in non-cases 34/73 (47) Median prednisone equivalent dose (IQR): In cases: 100 mg (89 to 129) vs. in non-cases: 107 (70 to 133) Anti IL-6: (Tocilizumab): In cases: 22/30 (73) vs. in non-cases 57/73 (78)	Azithromycin: In cases: 9/30 (30) vs. in non-cases: 31/78 (40)
Dellière [15]	Paris, France 15 March to 1 May 2020 4 ICUs of 2 university hospitals N = 366 (246 were mechanically ventilated). Analysis was performed on N = 108 (105 mechanically ventilated) who were clinically deteriorated and had >1 sample for mycology	Median age (IQR): 62 (56 to 68), 88% male (Baseline?) SOFA score, median (IQR): 6 (SD 3.8) Ventilation position: no information	10/108 (9.3)	Clinical worsening	Corticosteroids: 16/108 (14.8) (dexamethasone >1g): In cases: 6/21 (28.6) vs. in non-cases: 10 (11.5)	Azithromycin: In cases: 9/21 (42.9) vs. in non-cases: 17/87 (19.5)
Fekkar [11]	Paris, France. 6 March to 24 April 2020. Single-center (tertiary care). N = 260. Analysis was performed on N = 145 patients for fungal super infection	Median age (IQR): 55 (48 to 64), 72% male	22/143 (15.4) available information (14) Hematopoeitic stem cell allograft *n* = 1, Solid organ transplant: *n* = 14 Chronic steroid: *n* = 4	To confirm clinical suspicion of VAP or clinical worsening	Corticosteroids: 22/145 (16.7) Anti IL-6: (Tocilizumab and Sarilumab): 6/145 (6)	Azithromycin: 9/260 (3.4)
Gangneux [9]	Rennes, France No information regarding study period. Single-center (teaching hospital) N = 45 mechanically ventilated patients	Median age (IQR): 60 (53 to 71) years, 71.1% male Baseline SOFA score, median (IQR): 7 (2 to 11)	1/45 (2.2)	Standard respiratory samples every 2 weeks	No information	No information
Gonzalo Segrelles-Calvo [13]	Madrid, Spain. 1 February to 30 April 2020 Single-center (university hospital) N = 215 ICU patients	Median age (SD) of cases: 59.6 (15.2) years, 71.4% male SOFA score: no information Ventilation position: no information	0 in cases, no information on non-cases	Standard respiratory samples every 2 weeks	Corticosteroids: In cases: 4/7 (57.1) vs. in non-cases 187/208 (90) Anti IL-6: (Tocilizumab): In cases: 5/7 (71.4) vs. in non-cases 69/208 (33.3)	Azithromycin: In cases: 7/7 (100) vs. in non-cases: 194/208 (93.3)
Lewis White [14]	Wales No information regarding study period. ICUs across the country (no specific numbers mentioned) ** N = 257 ICU patients (135 of those were screened for invasive fungal disease, 122 were mechanically ventilated)	Median age: 57 (IQR) years (48 to 64), 2.2/1 male SOFA score: no information Ventilation position: no information	14/257 (10.4) Solid Cancer: *n* = 10, hematologic malignancies: 4	Samples were sent to the Public Health Wales Mycology reference laboratory at the discretion of clinicians	Corticosteroids: 32/57 (56.1) (no available data from the rest of the patients)	Antibiotics: 115/122 (94.3, no available data from the rest of the patients)
Razazi [10]	Creteil, France 1 October 2009 to 29 April 2020 * Single-center (tertiary hospital). N = 90 mechanically ventilated ICU patients >48 h	Median age (IQR) 59 (53 to 69), 82% male Baseline SOFA score, median (IQR): 7 (4 to 8) Prone position, *n* = 75 (83%)	14/90 (16) Solid cancer *n* = 5, hematologic malignancies *n* = 1, solid organ transplant *n* = 5, sickle cell, *n* = 3.	To confirm clinical suspicion of VAP	Corticosteroids: 35/87 (40.2, any dose) Anti IL-6: (Tocilizumab): 6/90 (6.7)	Antibiotics during first 24 h 90/90 (100)
Versyck [16]	Valenciennes, France. Between 15 March 2020 and 30 April 2020 Single center (general hospital). N = 54 mechanically ventilated ICU patients	Median age (range) 65 (44 to 83), 72% male Prone position, *n* = 34 (63)	Solid organ transplant *n* = 1, hematologic malignancies *n* = 1, chronic immunosuppression, *n* = 2.	When weekly tests of serum galactomannan and (1–3)-β-d-glucan were positive, BAL and chest CT scans were performed.	No information.	No information.

SD, standard deviation; IQR, interquartile range; ARDS, acute respiratory distress syndrome; ICU, intensive care unit * non-COVID-19 patients were included, ** Study type was descriptive study from a national Public Health Mycology Reference Center.

**Table 2 jcm-10-02469-t002:** Diagnostic methods applied to the patients and the definition of clinical entity used in the study, ordered by alphabetical order of first author’s names.

First Author (Ref) Number of Patients Used for Data Analysis	Radiology	Type of Respiratory Samples, n/N (%)	Mycology Tests in Respiratory Samples	Clinical Entity Used	Effect Size and Treatment	Time to Diagnosis of Clinical Entity
Bartoletti [12] N = 108, ICU population N = 163	No specific mention, but in CAPA and AspICU criteria, abnormal thoracic imaging is included.	BAL: 189 samples from 108 patients (median 2, IQR 2 to 5).	GM in BAL fluid ≥ 1: 30/108 (28%); on day 0–2: 14/108 (13%), and on subsequent time points: 9 (8%) and 5 (5%) *Aspergillus* culture positive in BAL fluid: 20/108 (18%) *Aspergillus* PCR positive in BAL: 26/67 performed (33.3%)	CAPA: 30 (27.7%) PIPA: 19 (17.6%)	CAPA vs. non-cases: Mortality: 44% vs. 19%, OR 3.5 (95%CI 1.4 to 9.7) PIPA vs. non-cases: Mortality: 74% vs. 26%, OR 11.6 (95%CI 3.2 to 41.3) 13/30 (43%) CAPA patients were treated with antifungal therapy, 46% died	Mean 4 days (SD 4) after admission to the ICU
Dellière [15] N = 108, ICU population 246	No specific mention, but in CAPA and EORTC/MSGERC criteria, pulmonary infiltrates and CT scan abnormalities were included.	BAL 80/108 (74.1%) ETA 22 (20.4%) Sputum 4 (3.7%) BA fluid 2 (1.9%)	GM in BAL positive: 8/19 cases (42.1%) *Aspergillus* culture positive respiratory sample: 17/19 cases (89.5%) *Aspergillus* PCR positive: 10/19 (52.6%)	CAPA Proven: *n* = 0 Probable: *n* = 19 (7.7% relative to all ventilated ICU patients)	CAPA vs. no CAPA: Mortality 71.4% vs. 36.8%, OR 4.3 (95%CI 1.5 to 12.1) * 6/19 CAPA patients were treated with antifungal therapy (all died) vs. 13/19 not treated (7 died)	Mean 6 days (SD 10) after admission to the ICU
Fekkar [11] N = 145, IC population 260	No specific mention, but in EORTC/MSG criteria, pulmonary infiltrates and CT scan abnormalities were included.	457 samples from 145 patients. BAL: 347 samples ETA: 120 samples DPS: 8 samples (mean 3 per patient).	GM in BAL positive: 3/7 cases (42.9%) *Aspergillus* culture-positive respiratory sample: 5/7 cases (71.4%) *Aspergillus* PCR positive: 4/7 (57%)	Putative IPMI EORTC/MSG: 7/260 (2.7%)	IPMI vs. no IPMI: Survival: 57% vs. 76%, OR 2.6 (95%CI 0.4 to 18.4, *p* = 0.2)	Median 7 days (IQR, 2 to 56) after admission to the ICU
Gangneux [9] N = 45	No specific mention, but in AspICU criteria, abnormal thoracic imaging is included.	BAL, BA, and ETA, but no information on numbers	No GM in BAL was performed. *Aspergillus* culture positive any respiratory samples: 9/45 (20%) *Aspergillus* PCR positive: 13/45 (28.9%)	AspICU: Putative: 9/45 (22%) Modified AspICU: Probable: 2/45 (7%) Putative: 4/45 (9%) Modified AspICU + positive PCR Putative: 15/45 (33.3%)	AspICU putative/probable vs. non-cases: Mortality: 28.6% vs. 13.3% All patients with putative/probable were treated with antifungal therapy (all died)	Not reported
Gonzalo Segrelles-Calvo [13] N = 215	RALE score severe group: in cases: 50%, vs. in non-cases: 46%	No information	Information only from cases: *Aspergillus* culture-positive: 6/7 BAL fluid, 2/7 BA, 1 serial sputum.	EORTC/MSG: *n* = 7	IFI EORTC vs. no IFI: Mortality: 86% vs. 37% (*p* < 0.05) 4/7 IFI EORTC/MSG patients were treated with antifungal therapy vs. 3/7 not treated (5/7 died)	Not reported
Lewis White [14] N = 135, ICU population 257	Cavitary lesion *n* = 5, nodules *n* = 5, tree in bud *n* = 1, others were not specific findings for COVID-19	BAL fluid (non- directed): 60/135 (44.4%)	GM in BAL fluid (no cut-off mentioned): 17/60 (28.3%) *Aspergillus* culture positive in any type respiratory samples: 11/135 (8.2%) *Aspergillus* PCR positive in BAL non-directed: 20/60 (33.3%) Serum BDG: 19/122 (15.6%)	AspICU: 8/135 (5.9%) IAPA: 20/135 (14.8%) CAPA: 19/135 (14.1%)	Patients with any mycology positive results vs. no positive results: 47.1% vs. 31.3% Mortality in: 5/8 AspICU, 9/20 IAPA, and 11/19 CAPA	Median 8 (range 0 to 35) after admission to the ICU
Razazi [10] N = 90	Pulmonary infiltrates: *n* = 90 (100%), cavitary lesion 1 (1.1%)	Available from: 58/90 (64.4) BAL fluid, *n* = 3 (5%) Protected telescope catheter sample, *n* = 55 (95%)	GM in BAL ≥ 1: not performed *Aspergillus* culture positive: 4/24 (16.7%) *Aspergillus* PCR positive: 16/81 (20%) Serum GM: no information	IAPA, Proven: *n* = 0 Probable: *n* = 7 (8%) Crude AspICU, Proven: *n* = 0 Putative: *n* = 2 (2%) Modified AspICU: Proven: 0 Putative: 6 (7%)	No analysis IAPA vs. no IAPA in COVID-19 patients. IAPA (including non COVID-19 patients) vs. no IAPA: Mortality ICU 58% vs. 34%, *p* = 0.02 Mortality day 28 50% vs. 33%, *p* = 0.11	Not reported
Versyck [16] N = 54	COVID-19 lesions on initial chest CT: <25% (11.1%), 25-50% (29.6%), 50-75% (38.9%), >75% (20.4%)	BAL 1/54 (1.9%) ETA 1/54 (1.9%)	GM in BAL ≥0.8: 1/1 (100%) *Aspergillus* culture positive: BAL 1/1 (100), ETA 1/1 (100)	IAPA, Probable: 2/54 (3.7%)	No analysis IPA vs. no IPA. Mortality in both IPA cases.	Mean 11 days (SD 6) after admission to the hospital

Abbreviations Types of tests: BDG, (1-3)-β-d-glucan; GM, galactomannan index. Types of respiratory samples: BA, bronchial aspirate; BAL, bronchoalveolar lavage; DPS, distal protected specimens; ETA, endotracheal aspirate. Other: RALE, Radiographic assessment of lungoedema Clinical entities; AspICU, *Aspergillus* in Intensive Care Unit; CAPA, COVID-19-associated pulmonary aspergillosis; IFI EORTC/MSG, invasive fungal infection according to EORTC/MSG; IPMI, invasive pulmonary mold infections EORTC/MSG; PIPA, putative invasive pulmonary aspergillosis (the criteria belong to these clinical entities are summarized in [21]. IQR, inter-quartile range; SD, standard deviation; ICU, intensive care unit * two additional patients with EORTC/MSGERC criteria were included in the analysis.

**Table 3 jcm-10-02469-t003:** Mycological findings.

First Author (Ref)	N ICU Patients	N of Patients BAL Fluid Available (%)	N Patients GM Positive in BAL (% Among Patients with Available BAL Fluid)	N Patients with Culture Positive in BAL (%)	N Patients PCR Positive in BAL (%)	N Patients Positive Serum GM (% Among Patients with Available Samples)	N of Cases According to Case Definition (%) ǂ
Bartoletti [12]	163	108 (66.3)	30 (27.7)	20 (18.5)	26/67 (38.8) ^d^	1/108 (0.9)	30 (18.4)
Dellière [15]	246	80 (32.5)	8 (10.0)	17 (15.7) ^c^	-	3/70 (4.3)	19 (7.7)
Fekkar [11]	260	145 (55.7)	11/333 (3.3) ^g^	255/474 (53.8) ^g^	17/449 (3.8) ^g^	3/503 (0.6) ^g^	7 (2.7)
Gangneux [9]	45	-	not performed	9 (20) ^c^	13/45 (28.8) ^c^	3/45 (6.7)	15 (33.3)
Gonzalo Segrelles-Calvo [13]	215	^≈^	^≈^	^≈^	^≈^	not performed	7 (3.3)
Lewis White [14]	257	60 (23.3) ^a^	17 (28.3)	11 (8.1) ^c^	20/60 (33.3)	19/122 (15.6) ^e^	20 (7.8)
Razazi [10]	90	58 (64.4) ^b^	not performed	4 (16.7) ^c^	16/81 (19.8) ^d^	5/88 (5.7)	7 (7.8)
Versyck [16]	54	1 (1.8) ^f^	1/1 (100)	1 (100)	-	2/54 (3.7)	2/54 (3.7)

ǂ: the highest number was mentioned here (% was re-calculated using number of ICU patients as denominator), -: no information or cannot be calculated,  ^≈^: information was available from cases only.^a^ BAL was performed through non directed bronchial lavage, ^b^ BAL only in 3 patients, in 55 patients procedure was through protected telescope, ^c^ % was calculated among all respiratory samples instead of among BAL samples only, ^d^ % was calculated among performed PCR in BAL samples (since PCR was not performed in all BAL samples), ^e^ no serum galactomannan was performed, (1-3)-β-d-glucannumber in serum/ plasma is presented here, ^f^ BAL was performed only if serum GM or (1-3)-β-d-glucan was positive, ^g^ instead of number of patients with BAL, details on number of positive samples from all respiratory samples were available. Grey square represents mycological finding that has the major impact on case definition in the mentioned study. Abbreviations: BAL, bronchoalveolar lavage; GM, galactomannan index; PCR, polymerase chain reaction.

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
