# Peer review of "Aspergillus in Critically Ill COVID-19 Patients: A Scoping Review"

_jcm, 2021, doi:10.3390/jcm10112469_

Round 1
Reviewer 1 Report
#JCM-1212371 - Review report
Manuscript: #1212371 - Aspergillus in critically ill COVID-19 patients: A Scoping Review
Date submitted: 22 April 2021
Deadline: 3 May 2021
In the manuscript entitled “Aspergillus in critically ill COVID-19 patients: A Scoping Review”, the authors designed a scoping review to clarify how the findings of COVID-19 associated pulmonary aspergillosis (CAPA) were interpreted. They first describe how they selected 123 publications and why they retained only 8. The quality of the selected studies was analyzed using modified Newcastle Ottawa score: 3 were judged as good, 2 of fair average quality, and 3 as poor. Extracted data, such as demographic and clinical characteristics of the patients included in the studies, are presented in Table 1, methods of diagnostic and definition of clinical entity are depicted in Table 2, and the different mycological criteria are described in Table 3. Finally, the authors identified 4 issues that are been discussed in the text: i) Diagnostic criteria based on mycological criterion. ii) Higher overall mortality associated with CAPA. iii) High mortality despite antifungal therapy. iv) Rare documented tissue invasion of Aspergillus.
Comments for the authors:
This review is interesting and well written but must be more structured to be clearer.
As the authors state, there have been numerous reports published on the scope of CAPA since the start of the COVID-19 pandemic and up to March 21st, 2021. The goals of this review were to clarify “how the findings of Aspergillus in COVID-19 patients were interpreted and whether it was associated with increased mortality” (abstract, line 15 on page 1, and line 53 on page 2) and “to find an explanation of the high incidence of positive Aspergillus tests” (line 61, on page 2). The study does not respond clearly to these two objectives such as it is presented here.
Page 8, line 3 it is stated that “Up until March 2021, no criteria had been agreed upon by relevant societies on the interpretation of positive Aspergillus test results in COVID-19 patients”, but on December, 14th, 2020, the European Confederation for Medical Mycology and the International Society for Human and Animal Mycology instituted a group of experts to propose consensus criteria for a case definition of CAPA and to provide up-to-date management recommendations for the diagnosis and treatment of patients with CAPA: Koehler, P. et al. "Defining and managing COVID-19-associated pulmonary aspergillosis: the 2020 ECMM/ISHAM consensus criteria for research and clinical guidance." The Lancet Infectious Diseases (2020). It is unexpected that a study based on the review of the literature does not refer to these guidelines, even if, in fine, it did not help clinicians during the first wave of the pandemic.
Minor points
- Do not use abbreviation in the abstract except if already defined (BAL, IPA).
- Overall, the manuscript is well written, but must be edited to correct some mistakes.
- Table 1 needs to be edited: frequencies were sometime indicated (x), sometime (x%), and sometime not indicated (e.g., ref 14: immunocompromised condition 4/54 (7.5) is missing). Several abbreviations were not explained (VAP, BAL, ICU).
- This study is based on the findings on CAPA and CAPA definition is done on page 14
- “We thank medical students Shereen Kadhum (SK), Sirin Yildirim (SY), and Cehdi Demir (CD)...” What is the participation of the persons listed as authors?
Major concerns:
- There is no conclusion in the different parts of the results section, so it is difficult to understand what is highlighted by this study.
- There is no conclusion in the different parts of the discussion section, so it is difficult to understand clearly what the message of the authors is.
- i) Diagnostic criteria based on mycological criterion.
- Page 14, line 4: Serum GM has limited utility because of its low sensitivity. In this context, screening with serum (1-3)-ß-D-glucan might show better performance characteristics as reported by White et al(12). This should be discussed.
- Page 14, line 8: “The prevalence of positive Aspergillus tests that were considered to be a case of IPA according to the criteria used by each of the individual studies ranged from 2.7% (9) to 33.3% (7)”. This observed heterogeneity should be mentioned and discussed in the light of the numerous factors that may contribute to the differing incidence rates: e.g., CAPA is difficult to diagnose particularly in patients presenting with severe COVID-19, in whom the clinical picture and radiological findings of ARDS resemble those of CAPA. Blood tests lack sensitivity due to the predominantly airway invasive growth of Aspergillus in non-neutropenic patients.
- ii) Higher overall mortality associated with CAPA.
- Page 14, line 13: We agree that the substantial higher mortality in patients with Aspergillus should be interpreted carefully (page 15, line 79) but the study (12) does not present this considerable disparity, maybe in relation to its very poor quality as it was shown in Sup. Table 3. This should be discussed.
- Page 14, line 32: “documented tissue invasion of Aspergillus was very rarely documented, as illustrated by the negative serum galactomannan test results in all but very few patients”. Serum GM is frequently negative in CAPA, including proven cases. Furthermore, a recent study by Flikweert et al., J Crit Care, 2020 did not associated high BAL GM index (>1) in blood with CAPA histopathological evidence.
- Page 14, line 38: “A critical look at how the patient samples were collected on which these tests were performed...”. Lower respiratory tract samples are the best specimens for fungal diagnostics, but their usage is reduced because of the risk of SARS-CoV-2 infection of health-care workers, and nosocomial transmission. This should be mentioned in the discussion.
- Page 15, line 78: What is the conclusion on the mycological test strategies?
iii) High mortality despite antifungal therapy.
- Page 15, line 58: “The authors of a proposed working definition of CAPA recognize this problem”. The reference is missing.
- Page 15, line 83: Voriconazole or isavuconazole are recommended as first-line treatment for possible, probable, and proven CAPA (Koehler et al).
- Page 15, line 87: Please cite the recruiting clinical trial: Isavuconazole for the Prevention of COVID-19-associated Pulmonary Aspergillosis (Isavu-CAPA) ClinicalTrials.gov Identifier: NCT04707703, and comment its interest in this field.
- iv) Rare documented tissue invasion of Aspergillus.
- Page 15, line 95: Other environmental factors related to the hospital overflow during the first wave of the pandemic, such as temporary facilities in hospitals or lack of rigorous ventilation requirements may also increase fungal exposure beyond what would normally be encountered within hospitals and/or intensive care units.
- Page 16, line 110: “Up to date, none of the published studies is originated from the second COVID-19 wave”. A better patients’ care according to diagnostic recommendations and anti-fungal treatment may be involved.
- Page 16, line 113: “...observational studies are mostly from France”. What should we understand?
Author Response
Reviewer #1
In the manuscript entitled “Aspergillus in critically ill COVID-19 patients: A Scoping Review”, the authors designed a scoping review to clarify how the findings of COVID-19 associated pulmonary aspergillosis (CAPA) were interpreted. They first describe how they selected 123 publications and why they retained only 8. The quality of the selected studies was analyzed using modified Newcastle Ottawa score: 3 were judged as good, 2 of fair average quality, and 3 as poor. Extracted data, such as demographic and clinical characteristics of the patients included in the studies, are presented in Table 1, methods of diagnostic and definition of clinical entity are depicted in Table 2, and the different mycological criteria are described in Table 3. Finally, the authors identified 4 issues that are been discussed in the text: i) Diagnostic criteria based on mycological criterion. ii) Higher overall mortality associated with CAPA. iii) High mortality despite antifungal therapy. iv) Rare documented tissue invasion of Aspergillus.
Comments for the authors:
This review is interesting and well written but must be more structured to be clearer.
>> Thank you for this comment and thank you for insightful and thoughtful additions that really improve our paper.
As the authors state, there have been numerous reports published on the scope of CAPA since the start of the COVID-19 pandemic and up to March 21st, 2021. The goals of this review were to clarify “how the findings of Aspergillus in COVID-19 patients were interpreted and whether it was associated with increased mortality” (abstract, line 15 on page 1, and line 53 on page 2) and “to find an explanation of the high incidence of positive Aspergillus tests” (line 61, on page 2). The study does not respond clearly to these two objectives such as it is presented here.
Page 8, line 3 it is stated that “Up until March 2021, no criteria had been agreed upon by relevant societies on the interpretation of positive Aspergillus test results in COVID-19 patients”, but on December, 14th, 2020, the European Confederation for Medical Mycology and the International Society for Human and Animal Mycology instituted a group of experts to propose consensus criteria for a case definition of CAPA and to provide up-to-date management recommendations for the diagnosis and treatment of patients with CAPA: Koehler, P. et al. "Defining and managing COVID-19-associated pulmonary aspergillosis: the 2020 ECMM/ISHAM consensus criteria for research and clinical guidance." The Lancet Infectious Diseases (2020). It is unexpected that a study based on the review of the literature does not refer to these guidelines, even if, in fine, it did not help clinicians during the first wave of the pandemic.
>> We have removed this sentence and included the paper mentioned by the reviewer in the revised version. In the mentioned paper, all criteria published to interpret positive Aspergillus findings are summarized, therefore we have removed Supplementary Table 4 and refer the criteria in interpreting the significance of Aspergillus findings to this paper.
Minor points
Do not use abbreviation in the abstract except if already defined (BAL, IPA).
>> we have written the abbreviations in full before using it later on tin the abstract.
Overall, the manuscript is well written, but must be edited to correct some mistakes.
Table 1 needs to be edited: frequencies were sometime indicated (x), sometime (x%), and sometime not indicated (e.g., ref 14: immunocompromised condition 4/54 (7.5) is missing). Several abbreviations were not explained (VAP, BAL, ICU).
>> Thank you for pointing this out. We have checked Table 1.
This study is based on the findings on CAPA and CAPA definition is done on page 14
“We thank medical students Shereen Kadhum (SK), Sirin Yildirim (SY), and Cehdi Demir (CD)...” What is the participation of the persons listed as authors?
>> The participation of the authors are as follow:
EY, LS, RASH, JPCvdA, LB, BJAR conceived the presented idea. EY drafted the article and performed data analysis, and EY, LB and BJAR interpreted the results of data analysis. EY, LB supervised medical students (SK, SY and CD) on performing data collection. EY, LS, RASH, JPCvdA, LB, BJAR discussed the results, gave critical revision, and gave final approval of the version to be published.
We have added this information in the acknowledgement section.
Major concerns:
There is no conclusion in the different parts of the results section, so it is difficult to understand what is highlighted by this study.
>> Our intention was to let the readers judge by themselves regarding the data since this is a scoping review. However, we have followed the suggestions by the reviewer by adding a summary to each results section.
There is no conclusion in the different parts of the discussion section, so it is difficult to understand clearly what the message of the authors is.
>> We have re-written several parts of the Discussion section to follow the advice from the reviewer.
- i) Diagnostic criteria based on mycological criterion.
Page 14, line 4: Serum GM has limited utility because of its low sensitivity. In this context, screening with serum (1-3)-ß-D-glucan might show better performance characteristics as reported by White et al(12). This should be discussed
>> We have discussed this issue in the revised version of the Discussion section (p. 16, line 112-116).
Page 14, line 8: “The prevalence of positive Aspergillus tests that were considered to be a case of IPA according to the criteria used by each of the individual studies ranged from 2.7% (9) to 33.3% (7)”. This observed heterogeneity should be mentioned and discussed in the light of the numerous factors that may contribute to the differing incidence rates: e.g., CAPA is difficult to diagnose particularly in patients presenting with severe COVID-19, in whom the clinical picture and radiological findings of ARDS resemble those of CAPA. Blood tests lack sensitivity due to the predominantly airway invasive growth of Aspergillus in non-neutropenic patients.
>> We have discussed this issue in the revised version of the Discussion section (p. 16, line 106-109). .
- ii) Higher overall mortality associated with CAPA.
Page 14, line 13: We agree that the substantial higher mortality in patients with Aspergillus should be interpreted carefully (page 15, line 79) but the study (12) does not present this considerable disparity, maybe in relation to its very poor quality as it was shown in Sup. Table 3. This should be discussed.
>> We have discussed this issue in the revised version of the Discussion section (p. 16, line 124-126).
Page 14, line 32: “documented tissue invasion of Aspergillus was very rarely documented, as illustrated by the negative serum galactomannan test results in all but very few patients”. Serum GM is frequently negative in CAPA, including proven cases. Furthermore, a recent study by Flikweert et al., J Crit Care, 2020 did not associated high BAL GM index (>1) in blood with CAPA histopathological evidence.
>> Thank you for this citation. We have added this information in the Discussion section (p. 16, line 116-119). .
Page 14, line 38: “A critical look at how the patient samples were collected on which these tests were performed...”. Lower respiratory tract samples are the best specimens for fungal diagnostics, but their usage is reduced because of the risk of SARS-CoV-2 infection of health-care workers, and nosocomial transmission. This should be mentioned in the discussion.
>> We have added this information (p.14 line 50-52)
Page 15, line 78: What is the conclusion on the mycological test strategies?
>> We have added several sentence regarding the mycological test strategies (p.15 line 90-94).
iii) High mortality despite antifungal therapy.
Page 15, line 58: “The authors of a proposed working definition of CAPA recognize this problem”. The reference is missing.
>> We have added the reference
Page 15, line 83: Voriconazole or isavuconazole are recommended as first-line treatment for possible, probable, and proven CAPA (Koehler et al).
>> We have added this information (p.17 line 179-180).
Page 15, line 87: Please cite the recruiting clinical trial: Isavuconazole for the Prevention of COVID-19-associated Pulmonary Aspergillosis (Isavu-CAPA) ClinicalTrials.gov Identifier: NCT04707703, and comment its interest in this field.
>> We have added this information (p.17 line 183-185).
- iv) Rare documented tissue invasion of Aspergillus.
Page 15, line 95: Other environmental factors related to the hospital overflow during the first wave of the pandemic, such as temporary facilities in hospitals or lack of rigorous ventilation requirements may also increase fungal exposure beyond what would normally be encountered within hospitals and/or intensive care units.
>> We have added this information (p.17 line 169-172).
Page 16, line 110: “Up to date, none of the published studies is originated from the second COVID-19 wave”. A better patients’ care according to diagnostic recommendations and anti-fungal treatment may be involved.
>> We have added this information(p.17 line 165-169).
Page 16, line 113: “...observational studies are mostly from France”. What should we understand?
>> There is a possibility that the Aspergillus testing, especially the use of galactomannan is more widespread in France ICU’s than in other countries. We have added this hypothesis.
Reviewer 2 Report
General comments
After several published reports of Aspergillus findings in COVID-19 patients led to the definition of a new disease entity- COVID-19 associated pulmonary aspergillosis, the authors designed this scoping review to clarify how the findings of Aspergillus spp. In COVID-19 patients were interpreted.
The main strength of this review is that it deals with a very interesting, highly topical and also very relevant clinical subject and overall, a lot of energy has gone into the work.
Specific comments
Title: OK
Introduction:
Line 41: word spacing
Line 46: word spacing
Material and Methods:
Line 62: word spacing
Results: OK
Page 8, Line 2: word-spacing
The Table 4 mentioned in the text is missing
References are not visible.
Author Response
Reviewer #2
General comments
After several published reports of Aspergillus findings in COVID-19 patients led to the definition of a new disease entity- COVID-19 associated pulmonary aspergillosis, the authors designed this scoping review to clarify how the findings of Aspergillus spp. In COVID-19 patients were interpreted.
The main strength of this review is that it deals with a very interesting, highly topical and also very relevant clinical subject and overall, a lot of energy has gone into the work.
>> Thank you for this comment.
Specific comments
Title: OK
Introduction:
Line 41: word spacing
>> corrected
Line 46: word spacing
>> corrected
Material and Methods:
Line 62: word spacing
>> corrected
Results: OK
Page 8, Line 2: word-spacing
>> corrected
The Table 4 mentioned in the text is missing
>> The table 4 were mentioned as Supplementary Table. Yet, we have removed supplementary Table 4 since a more extended information regarding criteria on interpreting positive aspergillus finding to define invasive pulmonary aspergillosis can be found in Koehler, P. et al. "Defining and managing COVID-19-associated pulmonary aspergillosis: the 2020 ECMM/ISHAM consensus criteria for research and clinical guidance." The Lancet Infectious Diseases (2020).
References are not visible.
>> I think there is something wrong with the reference sent by the publisher (also mentioned by reviewer #5. We have checked our version and the reference is visible. In the submitted version, we have checked it once again.
Reviewer 3 Report
Excellent Work, asking the Right Questions About the COVID associated pulmonary aspergillosis!
Is it only a marker or is it a really Problem. You deduced all the relevant Questions on it!
Very nice and good reading! Congratulations, in my opinion we have really to discuss CAPA in this way, as it is not clear if mortality is really attributable! So I think you will open the scientific discussion on this Topic! Bravo!
My only Suggestion: table 4 in the supplemental material should be integrated in the paper, as it is important to see on which Definition the different studies rely on.
Author Response
Reviewer #3
Excellent Work, asking the Right Questions About the COVID associated pulmonary aspergillosis!
Is it only a marker or is it a really Problem. You deduced all the relevant Questions on it!
Very nice and good reading! Congratulations, in my opinion we have really to discuss CAPA in this way, as it is not clear if mortality is really attributable! So I think you will open the scientific discussion on this Topic! Bravo!
>> Thank you for these compliments.
My only Suggestion: table 4 in the supplemental material should be integrated in the paper, as it is important to see on which Definition the different studies rely on.
>> Thank you for the suggestion. However, recently an extensive overview regarding various criteria in interpreting positive aspergillus finding to define invasive pulmonary aspergillosis has been published (Koehler, P. et al. "Defining and managing COVID-19-associated pulmonary aspergillosis: the 2020 ECMM/ISHAM consensus criteria for research and clinical guidance." The Lancet Infectious Diseases (2020)). Therefore, we have removed Supplementary Table 4 and referred to this publication as advised by reviewer #1.
Reviewer 4 Report
Thank you for giving me the opportunity to review this review on aspergillosis in COVID-19 patients. The paper is well written. I read this paper with great interest as aspergillosis in COVID-19 patients have gained interest with the increasing number of publications on the topic.
This review raises important issues concerning the diagnosis of aspergillosis and the high mortality rate found in the literature. The discussion is well written.
Why does the number of cases of aspergillosis appear high during this period ?
- Deos it results from increased screening of fungal markers in these patients ?
- Is it the use of therapies that interacts with the patient's immune system (Steroids, Eculizumab...) or
- Is it sepsis-induced immunoparalysis secondary to COVID-19 ?
Comments :
1) Do you know if the antigen was performed in each BAL ?
2) In the Table 3 was the number of GM positive in BAL normalised to the number of available GM in BAL ?
3) In the discussion you state :"CAPA is typically diagnosed in the second week after ICU admission and in a significant proportion even later..." I think this is an interesting data and I would suggest to add this information in a table. The onset of aspergillosis in these patients is a valuable information. Actually, the management of patients diagnosed with aspergillosis also depend on when aspergillosis is diagnosed during thier ICU stay. The maangement may differ if aspergillosis is suspected after 2 days of ICU or upon 3 weeks of intensive care.
4) Could a meta-analysis be conceivable ?
5) Recent papers are available in the literature and should probably be added in the references. An update of the review should be discussed according to the quality of the recent literature.
- Mitaka H, Kuno T, Takagi H, Patrawalla P. Incidence and Mortality of COVID-19-associated Pulmonary Aspergillosis: A Systematic Review and Meta-analysis. Mycoses. 2021 Apr 25. doi: 10.1111/myc.13292. Epub ahead of print. PMID: 33896063.
- Chong WH, Neu KP. The Incidence, Diagnosis, and Outcomes of COVID-19-associated Pulmonary Aspergillosis (CAPA): A Systematic Review. J Hosp Infect. 2021 Apr 20:S0195-6701(21)00163-8. doi: 10.1016/j.jhin.2021.04.012. Epub ahead of print. PMID: 33891985; PMCID: PMC8057923.
- Lahmer T, Kriescher S, Herner A, Rothe K, Spinner CD, Schneider J, Mayer U, Neuenhahn M, Hoffmann D, Geisler F, Heim M, Schneider G, Schmid RM, Huber W, Rasch S. Invasive pulmonary aspergillosis in critically ill patients with severe COVID-19 pneumonia: Results from the prospective AspCOVID-19 study. PLoS One. 2021 Mar 17;16(3):e0238825. doi: 10.1371/journal.pone.0238825. PMID: 33730058; PMCID: PMC7968651.
Minor corrections :
1) Table 2 beta-diglucan --> beta-D-glucan
2) Page 12 line 31 :
instead proportion 31 among patients --> instead of the
3) Page 14 line 4
4) 1,3 beta-diglucan --> beta-D-glucan
5) supplementary table 5 : line Elsoukkary (15) : please add in the remark box : n.a. for homogeneity
6) In the text you have 21 references. However, I can only see 19 in the reference section.
Author Response
Thank you for giving me the opportunity to review this review on aspergillosis in COVID-19 patients. The paper is well written. I read this paper with great interest as aspergillosis in COVID-19 patients have gained interest with the increasing number of publications on the topic.
This review raises important issues concerning the diagnosis of aspergillosis and the high mortality rate found in the literature. The discussion is well written.
>> Thank you for this compliment.
Why does the number of cases of aspergillosis appear high during this period ? Does it results from increased screening of fungal markers in these patients ? Is it the use of therapies that interacts with the patient's immune system (Steroids, Eculizumab...) or, Is it sepsis-induced immunoparalysis secondary to COVID-19 ?
>> Thank you for this insight. We have added an paragraph regarding these hypotheses.
Comments :
1) Do you know if the antigen was performed in each BAL ?
>> It is not always possible to derive the number of BAL performed from each patient and whether antigen test was performed in every BAL. It depended also on the study. It is clear for example that in the study Razazi, et al. and Gangneux, et al that no antigen test in BAL was performed. In a study antigen determination information in BAL was only available in cases and not in other study population. In some studies, antigen tests seemed to be performed in all BAL (for example Bartoletti et al). Information on the number of BAL samples is presented in Table 2 and in Table 3
2) In the Table 3 was the number of GM positive in BAL normalised to the number of available GM in BAL ?
>> The number of GM positive in BAL is Table 3 is normalized to the number of patients with with available BAL fluid, and the number of patient with at least 1 BAL fluid is presented in Table 2. We have revised the heading of Table 3 to make this information more clear.
3) In the discussion you state :"CAPA is typically diagnosed in the second week after ICU admission and in a significant proportion even later..." I think this is an interesting data and I would suggest to add this information in a table. The onset of aspergillosis in these patients is a valuable information. Actually, the management of patients diagnosed with aspergillosis also depend on when aspergillosis is diagnosed during thier ICU stay. The maangement may differ if aspergillosis is suspected after 2 days of ICU or upon 3 weeks of intensive care.
>> We have included this information in Table 2.
4) Could a meta-analysis be conceivable ?
>> Theoretically, meta-analysis or a systematic review is possible. This has been performed too as this reviewer mentioned below. However, they all having starting point the clinical entity COVID-19 associated pulmonary aspergillosis. Yet, in clinical practice, it is still doubtful whether this entity exist (as experienced by the authors and agreed by several reviewers here), and this scoping review differs from these published systematic reviews because it aimed to explore how do the authors come to the conclusion of this clinical entity.
5) Recent papers are available in the literature and should probably be added in the references. An update of the review should be discussed according to the quality of the recent literature.
Mitaka H, Kuno T, Takagi H, Patrawalla P. Incidence and Mortality of COVID-19-associated Pulmonary Aspergillosis: A Systematic Review and Meta-analysis. Mycoses. 2021 Apr 25. doi: 10.1111/myc.13292. Epub ahead of print. PMID: 33896063.
Chong WH, Neu KP. The Incidence, Diagnosis, and Outcomes of COVID-19-associated Pulmonary Aspergillosis (CAPA): A Systematic Review. J Hosp Infect. 2021 Apr 20:S0195-6701(21)00163-8. doi: 10.1016/j.jhin.2021.04.012. Epub ahead of print. PMID: 33891985; PMCID: PMC8057923.
Lahmer T, Kriescher S, Herner A, Rothe K, Spinner CD, Schneider J, Mayer U, Neuenhahn M, Hoffmann D, Geisler F, Heim M, Schneider G, Schmid RM, Huber W, Rasch S. Invasive pulmonary aspergillosis in critically ill patients with severe COVID-19 pneumonia: Results from the prospective AspCOVID-19 study. PLoS One. 2021 Mar 17;16(3):e0238825. doi: 10.1371/journal.pone.0238825. PMID: 33730058; PMCID: PMC7968651.
>> Thank you for this addition. We have mentioned the first two papers in our present manuscript on how our scoping review differs from these two reviews in the Introduction part. We have looked at the study from Lahmer and colleague but it does not fulfil our inclusion criteria at the initiation of this review (i.e. >35 patients).
Minor corrections :
1) Table 2 beta-diglucan --> beta-D-glucan
>> we have corrected this
2) Page 12 line 31 : instead proportion 31 among patients --> instead of the
>> we have corrected this
3) Page 14 line 4 1,3 beta-diglucan --> beta-D-glucan
>> we have corrected this
5) supplementary table 5 : line Elsoukkary (15) : please add in the remark box : n.a. for homogeneity
>> we have removed supplementary Table 5 because it was accidentally added.
6) In the text you have 21 references. However, I can only see 19 in the reference section.
>> There are two reference lists, one for the manuscript (22 references) and one for the supplementary tables (19 references). In the revised version, the numbers are changed, and we have checked the reference carefully.
Reviewer 5 Report
General consideration:
- Problems with references in the table 1. In the column “first author”, the number (ref) is not related to the bibliography. I think we can find this problem everywhere in all tables of the paper as well in the supplement tables. One of the most important problems is that for example Bartoletti (ref10) does not exist anywhere? This article does not appear in the bibliography. Authors have to control and complete the references and bibliography list.
- In the bibliography ref 1: the journal, date, page are missing. This reference #1 appears as Bartoletti in the supplementary table 3. It is nor correct. There are many other examples.
- Conclusion of the abstract is not the same message as in the article. Abstract: Concluding, this review showed how studies reported Aspergillus findings and linked it with clinical outcomes in COVID-19. Text: In conclusion, in this scoping review, we identified several factors that should be considered in determining the significance of Aspergillus spp. findings in respiratory materials of COVID-19 patients, for clinical and research purposes.
I think the message is not exactly the same. For clinicians, the importance of biological materials could be emphasized. And that there are many differences between centers in terms of biomarker use/availability/choice.
4. Line 107, 108, 110: bibliography references not correct.
5. Nothing about alternatives treatment for Anti SARS-CoV-2 as tocilizumab, antiviral therapy… It could be interesting to know if patients received other treatment than corticosteroids. The role of tocilizumab can be questioned. Some studies show this details
6. Razazi et al had collected patients with ARDS since 2009. In Table 1, the authors could have differentiated patients with SARS-CoV-2 infection from patients with other viral infections.
7. Line 107, the reference #7 is not correct. 45 pts included in the Gagneux et al. study (Ref 4 in the additional materials)
8. Table 1 page 6/16. Study of Gonzalo et al.: right ref is the 5.
9. Table 1 page 7/16: Razazi et al. Right Ref is 7. In the column of corticosteroids use, it’s written 3 patients did not have ARDS: but in the article the C-ARDS are 90 (11 mild, 49 moderate, 30 severe), all ARDS! I think that the authors could leave this sentence.
10. Table 1 page 7/16: Versyck et al : right ref is 8 and not 14. In the column of settings: March 15th 2020 to…
11. Table 1 page 9/16: Bartoletti et al is not mentioned in the bibliography !!! Those authors made some comments about other treatments against COVID as for example tocilizumab (>75% received this drug)
12. Table 2 page 9/16: First column: IC is not defined. Does it mean ICU or something other? For Dellière et al. : IC ? and wrong reference too.
13. Table 2 page 9/16: PIPA is not defined
14. Table 2 page 9/16: Aspergillus PCR positive 10/19 (20%): percentage is not correct
15. Table 2 page 11/16: Versyck et al. Wrong #ref
16. Table 3 : all references to correct
17. Line 122 page 16/16 : “… to a false negative autopsy but we… “ and not I.
Comments to the authors
This review is very interesting regarding the first wave of COVID. Perhaps table 1 could be simplified. Do we need really the details about the immunocompromised conditions?
Some comments about COVID treatment may give a supplementary view on the differences between the centers. Discussion about antifungal treatment: some words on preemptive, empirical or directed antifungal treatments could be interesting but not mandatory.
You and your students have done a very good job!
Author Response
Reviewer#5
General consideration:
- Problems with references in the table 1. In the column “first author”, the number (ref) is not related to the bibliography. I think we can find this problem everywhere in all tables of the paper as well in the supplement tables. One of the most important problems is that for example Bartoletti (ref10) does not exist anywhere? This article does not appear in the bibliography. Authors have to control and complete the references and bibliography list.
>> We have checked in the submitted version and we do have this reference. Also, we have checked for all other comments regarding references below, we do have the correct references in the version we submitted. Is there perhaps a problem from the submitting site? We have experienced this problem too before. The reviewers received the paper in MS Word, unlike many other journals in PDF. In MS Word change on the bibliography can happen accidentally.
- In the bibliography ref 1: the journal, date, page are missing. This reference #1 appears as Bartoletti in the supplementary table 3. It is nor correct. There F are many examples.
>> Regarding reference#1 in the Supplementary Table, it Is indded that we put a wrong reference. We have corrected this.
- Conclusion of the abstract is not the same message as in the article. Abstract: Concluding, this review showed how studies reported Aspergillus findings and linked it with clinical outcomes in COVID-19. Text: In conclusion, in this scoping review, we identified several factors that should be considered in determining the significance of Aspergillus spp. findings in respiratory materials of COVID-19 patients, for clinical and research purposes.
I think the message is not exactly the same. For clinicians, the importance of biological materials could be emphasized. And that there are many differences between centers in terms of biomarker use/availability/choice.
>> We have revised the abstract according to your input. Thank you.
- Line 107, 108, 110: bibliography references not correct.
>> We have checked the reference once again and in our version this is correct.
- Nothing about alternatives treatment for Anti SARS-CoV-2 as tocilizumab, antiviral therapy… It could be interesting to know if patients received other treatment than corticosteroids. The role of tocilizumab can be questioned. Some studies show this details
>> Thank you for this question. We have added the information on tocilizumab in Table 1.
- Razazi et al had collected patients with ARDS since 2009. In Table 1, the authors could have differentiated patients with SARS-CoV-2 infection from patients with other viral infections.
>> Thank you for pointing this out. We have added several sentences regarding this comparison between ARDS in COVID an ARDS due to other viruses in the Discussion section (p.17, line 160-165).
- Line 107, the reference #7 is not correct. 45 pts included in the Gagneux et al. study (Ref 4 in the additional materials)
>> We have checked the reference once again and in our version this is correct.
- Table 1 page 6/16. Study of Gonzalo et al.: right ref is the 5.
>> We have checked the reference once again and in our version this is correct.
- Table 1 page 7/16: Razazi et al. Right Ref is 7. In the column of corticosteroids use, it’s written 3 spatients did not have ARDS: but in the article the C-ARDS are 90 (11 mild, 49 moderate, 30 severe), all ARDS! I think that the authors could leave this sentence.
>> We have checked the reference once again and in our version this is correct. We have removed the sentence.
- Table 1 page 7/16: Versyck et al : right ref is 8 and not 14. In the column of settings: March 15th 2020 to…
>> In our version the reference is correct. There are reference lists. One for the manuscript and the other is for the supplementary table. The duration of the study from Versyck, and colleagues was also already mentioned in the submitted version.
- Table 1 page 9/16: Bartoletti et al is not mentioned in the bibliography !!! Those authors made some comments about other treatments against COVID as for example tocilizumab (>75% received this drug)
>> We have checked in the submitted version and we do have this reference. Also, we have checked for all other comments regarding references below, we do have the correct references in the version we submitted. Is there perhaps a problem from the submitting site?
- Table 2 page 9/16: First column: IC is not defined. Does it mean ICU or something other? For Dellière et al. : IC ? and wrong reference too.
>> Thank you for pointing this out. It should be ICU instead of IC. We have corrected it. Regarding the reference, in our version, we do have correct reference.
- Table 2 page 9/16: PIPA is not defined
>> We have added the abbreviation of PIPA
- Table 2 page 9/16: Aspergillus PCR positive 10/19 (20%): percentage is not correct
>> Thank you for pointing this out. This has been corrected.
- Table 2 page 11/16: Versyck et al. Wrong #ref
>> Regarding the reference, in our version, we do have correct reference.
- Table 3 : all references to correct
>> We have checked the reference once again and in our version this is correct
- Line 122 page 16/16 : “… to a false negative autopsy but we… “ and not I.
>> we have corrected this.
Comments to the authors
This review is very interesting regarding the first wave of COVID. Perhaps table 1 could be simplified. Do we need really the details about the immunocompromised conditions?
>> Thank you for this advice. However, we are convinced that we need to keep this details since one of the risk factors in having invasive pulmonary aspergillosis is the immunocompromised status such as in patients with hematologic malignancies.
Some comments about COVID treatment may give a supplementary view on the differences between the centers. Discussion about antifungal treatment: some words on preemptive, empirical or directed antifungal treatments could be interesting but not mandatory.
>> We have added this information in the Discussion section.
You and your students have done a very good job!
>> Thank you for this compliment